# Adsorption of Methylene Blue on the Surface of Polymer Membrane; Dependence on the Isotopic Composition of Liquid Matrix

**DOI:** 10.3390/polym14194007

**Published:** 2022-09-25

**Authors:** Nikolai F. Bunkin, Polina N. Bolotskova, Yana V. Gladysheva, Valeriy A. Kozlov, Svetlana L. Timchenko

**Affiliations:** Department of Fundamental Sciences, Bauman Moscow State Technical University, 2-nd Baumanskaya Street 5, 105005 Moscow, Russia

**Keywords:** fuel cells, spectrophotometry, polymer membrane, Nafion, electrically charged polymer surface, methylene blue, adsorption, desorption of water, diffusion processes, deuterium-depleted water, Lennard–Jones equation

## Abstract

As was found in our previous works, when Nafion swells in water, polymer fibers unwind into the bulk of the surrounding liquid. This effect is controlled by the content of deuterium in water. Here, we present the results of studying the dynamics of methylene blue (MB) adsorption on the Nafion surface for MB solutions based on natural water (deuterium content is 157 ppm, the unwinding effect occurs) and based on deuterium-depleted water (DDW; deuterium content is 3 ppm, there is no unwinding). In addition, we studied the dynamics of water desorption during drying of the Nafion polymer membrane after soaking in MB solution based on natural water and DDW. It turned out that in the case of natural water, the rate of MB adsorption and water desorption is higher than in the case of DDW. It also turned out that the amount of MB adsorbed on the membrane in the case of natural water is greater than in the case of DDW. Finally, it was found that the desorption of water during drying is accompanied by a rearrangement of the absorption spectrum of Nafion. This rearrangement occurs earlier in the case of DDW. Thus, by infinitesimal changes in the deuterium content (from 3 to 157 ppm) in an aqueous solution, in which a polymer membrane swells, we can control the dynamics of adsorption and desorption processes. A qualitative model, which connects the observed effects with the slowing down of diffusion processes inside the layer of unwound fibers, is proposed.

## 1. Introduction

The specific features of water interaction with hydrophobic and hydrophilic polymer membranes are currently a hot topic in various fields of physical chemistry. Just as challenging is the physical nature of the nanostructures that are formed inside the polymer matrix. For a background on Nafion™ and its properties, see the review in Ref. [1].

Nafion (C_7_HF_13_O_5_S × C_2_F_4_) is manufactured via the copolymerization of a perfluorinated vinyl ether comonomer with tetrafluoroethylene, resulting in the chemical structure given below:



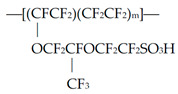



Nafion is composed of perfluoro-vinyl ether groups terminated with sulfonic groups on a tetrafluoroethylene (Teflon) backbone. Teflon is a highly hydrophobic matrix, while the sulfonic groups are very hydrophilic moieties. Therefore, when Nafion is soaked in water, the possibility of direct and reverse micellization appears (see [1]). Such a possibility is of great interest from a fundamental viewpoint, and also opens up a number of interesting applications in various industrial fields.

The Nafion matrix is biocompatible and flexible, and it has excellent mechanical and chemical stability. When studying the specific features of Nafion, which are manifested during soaking in water, it is necessary to take into account the dissociation of terminal sulfonic groups at the polymer–water interface. The chemical formula of this reaction is R—SO_3_H + H_2_O ⇔ R—SO_3_^−^ + H_3_O^+^. In addition, it is necessary to take into account the formation of through channels with a diameter of 2–3 nm in the bulk of the polymer matrix (see [1] for more details). Negatively charged areas at the inner surface of these channels provide a possibility for cations to transit through the membrane bulk. Furthermore, the nanometer-sized structure of these channels allows for the separation of H^+^ and OH^−^ ions from both sides of the membrane, which is used in low-temperature hydrogen fuel cells. The physical mechanism of such a separation has been comprehensively studied; see, for example, the review [2]. Note also that, currently, the anion-exchange membranes are also widely explored; see a recent review [3].

Various technologies for the introduction of certain external particles into polymer matrices have been actively developed. The reason is to improve the characteristic properties of the membrane. For example, some particles, interacting with polymer matrices, act as plasticizers and depress the polymer’s glass transition temperature; see Ref. [4]. In some cases, the oxides of various metals are embedded into the bulk of the polymer to promote the generation of reactive oxygen species (ROS) inside the membrane; see, for example, studies [5,6,7]. The effect of ROS generation is widely applied in agriculture, specifically in greenhouses based on polymer films for the destruction of bacterial cell walls and their subsequent death; see, for example, [8].

However, ROS is a parasitic factor for polymeric membranes as a fuel cell element. Indeed, an important feature of Nafion-based proton exchange membrane fuel cells (PEMFCs) is the chemical degradation of the membrane due to the oxidative attack of the polymer chains by ROS that are generated during fuel cell operation. The problem of increasing the durability of the polymer membrane as a fuel cell element while suppressing the membrane degradation is very clear. This can be achieved, for example, by adsorption of various antioxidants on the membrane surface; see recent works [9,10].

In this work, we studied the adsorption of an antioxidant based on methylene blue on the surface of Nafion. Methylene blue (MB) or methylthionine chloride, C_16_H_18_ClN_3_S, is a tricyclic phenothiazine dye that is deep blue in color; antioxidant properties of MB were described in detail in a recent work [11]. MB undergoes reduction by nicotinamide adenine dinucleotide phosphate to produce leucomethylene blue (leucoMB), which is a colorless compound. The choice of this particular antioxidant is due to the following. As is known [12], MB in aqueous solutions is a Zn^+2^ ionophore, i.e., it is able to transfer this ion across a lipid membrane in a living cell. At the same time, Nafion is also capable of passing cations of certain sizes through the pores formed during swelling in water (see Ref. [1]), i.e., the properties of Nafion and MB in water appear to be close to one another.

As is known [12], the Zn^2+^ ions inhibit the elongation of ribonucleic acid (RNA) polymerase, a component of the RNA viruses that are not found in the human body. In fact, MB has the ability to transport Zn^2+^ across the viral envelope by endo-lysosomes. Therefore, MB is widely used to treat patients with SARS-CoV-2; see [13,14,15,16,17].

Thus, the study of the interaction between the Nafion polymer membrane and MB is interesting not only from the viewpoint of studying the possibility of increasing the durability of the polymer membrane in the fuel cell, but also in the context of biophysical applications. Indeed, as was previously shown in our works, devoted to the study of Nafion membrane swelling in aqueous media with different deuterium contents [18,19,20], when swelling in natural water (the deuterium content in this liquid is 157 ppm, see [21]), the polymeric fibers unwind toward the water bulk. These fibers do not completely tear off the polymer substrate, forming a flexible brush-type structure, and the size of this structure in the bulk of water amounts to 300 microns [19]. At the same time, when the membrane is soaked in deuterium-depleted water (DDW; the deuterium content is 3 ppm), the effect of unwinding is absent.

As was concluded in works [18,19,20], a polymer membrane containing through channels inside its matrix and decorated with unwound polymer fibers is similar to a cell membrane. Indeed, the cell membrane also contains through channels inside the lipid bilayer, and is surrounded from the outside by the polysaccharide fibers (glycocalyx, or extracellular matrix, ECM); see [22]. Obviously, this analogy should be considered only taking into account the difference in spatial scales. It is very important that ECM is one of the key elements within a solid tumor that restricts drug penetration into tumor cells. This aspect of tumor biology was recognized for decades and has been extensively reviewed [23,24,25,26] as have efforts to remove the ECM with enzymes or collagen/hyaluronan synthesis inhibitors. Therefore, studies of the interaction of a polymer membrane with polymer fibers unwound into a liquid volume with antioxidants of MB type are interesting from the viewpoint of modeling similar processes on cell membranes.

We could find in the literature only one article [27], devoted to the Nafion membrane containing MB nanoparticles. Apparently, this work should be considered as the first study of the properties of Nafion modified by MB. An important feature of our work is that we studied the dynamics of MB adsorption on the Nafion surface from aqueous solutions of MB, which were prepared on the basis of natural water and DDW. Thus, in this experimental protocol, it becomes possible for the first time to study the dynamics of adsorption on the membrane surface, taking into account the effect of unwinding of polymer fibers. Concluding this section, we note that in our experiment, we used a spectroscopic technique based on measuring the time dependence of the absorption coefficient of an aqueous solution of MB during adsorption. This technique was used in recent works [28,29,30,31,32], where the dynamics of MB adsorption from water and aqueous solutions on the surface of various adsorbents was explored.

## 2. Materials and Methods

The Nafion N117 plates (Sigma Aldrich, St. Louis, MO, USA) with a thickness of 175 μm and an area of *S* = 7.5 cm^2^ were investigated. The test liquids were deionized (natural) water (deuterium content is 157 ± 1 ppm) with a resistivity of 18 MΩ × cm at 25 °C, refined by a Milli-Q apparatus (Merck KGaA, Darmstadt, Germany), and deuterium-depleted water (DDW, deuterium content is 1 ppm), purchased from Sigma Aldrich, St. Louis, MO, USA. For the preparation of MB solutions with different weight concentrations, ultrahigh-purity methylene blue (MB) powder was purchased from the manufacturer Macsen Labs (N.K. Agrawal Group, Udaipur, Rajasthan, India).

Experiments were carried out according to two protocols. According to the first protocol, the Nafion plate was soaked for a certain time (in our case, this time was 80 min) in a Petri dish, containing 12 mL of aqueous solution of MB with different weight concentrations. While soaking, MB was adsorbed on the Nafion surface, and the membrane acquired a blue color, i.e., the color of the initial MB solution. In this case, a discoloration (dye removal) of the MB solution was observed. While MB is being adsorbed on the Nafion surface, a portion of the MB solution (with volume of 2 mL) was taken from the Petri dish for spectrophotometric measurements. Then, we recorded the absorption spectrum of the MB solution in the spectral range of 190–900 nm. Then, this portion of the MB solution was again poured into the Petri dish, in which the Nafion plate continued soaking. During this experiment, the Petri dish was sealed, and the evaporation of liquid can be neglected. The process of discoloration of the MB solution could be monitored visually; see Figure 1.

According to the second protocol, after soaking the Nafion plate in the MB solution, the absorption spectrum of the plate during its drying under the laboratory conditions (temperature *T* = 25 °C; relative humidity RH = 50%) was explored. To do that, the Nafion plate was placed inside the spectrophotometer, and the absorption spectrum of the plate was recorded with a certain time interval. As the water content in the near-surface layer of the membrane decreases upon drying, the effective density of the MB contained in this layer increases, and the blue color of the plate becomes deeper.

The spectroscopic experiments were carried out using a GBC Cintra 4040 Spectrophotometer (UVISON Technologies Limited, Kent, UK) with a dual Littrow monochromator; the spectral range is 190–900 nm. In this device, the slit width can be changed in the range from 0.1 to 2.0 mm, and the presence of a double Littrow monochromator in the Czerny–Turner configuration guarantees high sensitivity, low stray light, background, and baseline drift.

In Figure 2, we show a graph of the absorption coefficient of the original dry Nafion plate N117. Note that this graph almost exactly repeats the dependences obtained for Nafion in Ref. [33].

As noted above, the MB solutions were prepared on the basis of natural water and DDW with different weight concentrations. In Figure 3, we show the absorptivity spectra of the MB solution with a concentration of 0.015 mg/mL, based on natural water and DDW. As follows from the graphs, the absorption spectra of both solutions are slightly different. Following the experimental protocol described in [28,29,30,31,32], we studied the temporal dynamics of the absorption spectra at the longest wavelength λ = 650 nm.

In Figure 4, we present the absorption spectra of two Nafion plates, which were soaked in the MB solutions with a weight concentration of 0.015 mg/mL based on natural water and DDW; the graphs were obtained immediately after taking the Nafion plate out of liquid. As follows from the graphs in Figure 2, Figure 3 and Figure 4, the spectra of the initial Nafion and Nafion, soaked in the MB solutions based on ordinary water and DDW, are different. In fact, new absorption bands appear in the process of soaking, which was previously obtained in [27]. Furthermore, the absorption spectra of Nafion, recorded immediately after removing the Nafion plate from the MB solutions, differ from the spectra of these solutions. As seen in Figure 3 and Figure 4, the low-frequency absorption band has the form of a doublet, and the central peaks of this doublet for the MB solutions are centered at wavelengths λ = 610 and 650 nm (see Figure 3), while in the case of the Nafion plate, which was soaked in these solutions, these peaks are redshifted and become equal, respectively, to λ = 641 and 736 nm. This can be associated with the formation of Coulomb and van der Waals molecular complexes, which appear due to physical adsorption; chemical adsorption, which resulted from the formation of chemical bonds between the adsorbed substance and the adsorbent, most likely does not occur, because of the chemical inertness of the Nafion fluorocarbon base [1].

Assuming that the absorption bands are associated with the electronic-vibrational state of molecules, the redshift in these bands means an increase in the effective mass of the oscillator. Indeed, the oscillator frequency is $\omega =\sqrt{\frac{k}{m}},$ where *k* is the elasticity constant, which stands for the attraction between the oscillator with effective mass *m* and the attracting center. Unfortunately, we are not yet able to offer a more detailed analysis of the redshift in the absorption spectrum.

As noted above, the evolution of the MB solutions spectra was investigated by recording the spectral pattern in its maximum at wavelength λ = 650 nm vs. the soaking time *t*. In the soaked Nafion plate, this maximum is redshifted apparently due to the formation of new complexes based on Nafion, water, and MB; this maximum is centered at wavelength λ = 736 nm, cf., Figure 3 and Figure 4. It seems reasonable that we should investigate the dynamics of drying of the Nafion plate by recording the Nafion absorptivity at wavelength λ = 736 nm.

In the next section, we present the results on the dynamics of discoloration (dye removal) of aqueous MB solutions based on natural water and DDW upon soaking of the Nafion plate in these solutions, as well as on the dynamics of drying of this plate.

## 3. Experimental Results

In Figure 5a, we show the absorption spectra of the MB solution with a weight concentration of 0.015 mg/mL on the basis of natural water (deuterium content is 157 ppm) and recorded upon soaking. In Figure 5b, we show the absorption spectra of the MB solution of the same concentration on the basis of DDW (deuterium content 1 ppm). Similar dependences were obtained for concentrations 0.02 and 0.025 mg/mL (not shown in Figure 5). As already noted, MB particles are adsorbed on the membrane surface, and therefore, the solution becomes lighter as Nafion is being soaked, i.e., we actually study the kinetics of MB removal from liquid. As follows from the graphs, the absorption coefficient for all spectral lines decreases upon soaking. For a quantitative analysis of the adsorption kinetics, we chose the line centered at wavelength λ = 650 nm (marked with a dashed line); see above. We denote the absorption coefficient at this wavelength as κ.

The graphs in Figure 6 show the dependences {[κ(*t* = 0) − κ(*t*)]/κ(*t* = 0)} × 100% for MB concentrations of 0.015, 0.02, and 0.025 mg/mL. The obtained dependences can be interpreted as a percentage of the MB amount, removed from the MB solution during adsorption. Panel (a) corresponds to the solution based on natural water, and panel (b) corresponds to the solution based on DDW.

In Figure 7a,b, we show the temporal dependence of Nafion absorptivity upon drying. Here, we investigated the spectral patterns of Nafion plates, soaked for 80 min in the MB solution with a concentration of 0.015 mg/mL, based on natural water (panel (a)) and DDW (panel (b)). Similar dependences were obtained for concentrations 0.02 and 0.025 mg/mL. We do not show the dependences for these concentrations, as they are qualitatively similar to the dependences for 0.015 mg/mL. The curves taken at the time *t* = 0 correspond to the spectral dependences shown in Figure 4, i.e., these measurements were made immediately after removing the Nafion plate from the liquid.

As follows from these graphs, at the 12th minute of drying the Nafion plate, which was soaked in the solution based on natural water, the initially doublet band turns into a triplet; see the pink graph on panel (a) in Figure 7. Upon further drying, we obtain once again a doublet structure for this band, but the high-frequency line of this doublet is redshifted. The same triplet–doublet transition occurs already at the 4th minute of drying after soaking in the solution based on DDW; see the red graph in panel (b). The triplet structures are especially illustrated in Figure 8a,b. Panel (a) is related to soaking in the MB solution based on natural water, *t* = 12 min; panel (b) is related to soaking in the solution based on DDW, *t* = 4 min.

As noted in Section 2, we studied the dynamics of Nafion drying by recording the absorptivity at wavelength λ = 736 nm vs. the time of drying. We denote this absorptivity as κ‘. In Figure 9a, we show the dependences κ‘(*t*) for the solutions based on natural water and DDW, and in Figure 10b, we show the dependences *d*κ‘/*dt* for these solutions.

## 4. Discussion

The results, shown in Figure 6, can be interpreted in terms of the diffusion-limited model of adsorption; see [34] and the review [35]. Namely, in [34], the equation
(1)$$\frac{dn}{dt}=C_{b}\left(C_{S_{0}}-n\right)\sigma k_{a}-nk_{d}$$
was solved. Here, *n* is the number of adsorbed molecules on the surface unit, *C_b_* is the concentration of adsorbed substance in the bulk of the solution, *C*_S0_ is the number of free adsorption sites per the surface unit before the beginning of the adsorption process (*C*_S0_ is a constant value), *σ* is the area of adsorbed molecules, and *k_a_* and *k_d_* are two proportionality constants. Obviously, *C_b_* = *C_b_*(*t*) is decreasing as a function of time. The solution to Equation (1) at the initial condition *n* = 0 for *t* = 0 has the following form (see [34]):(2)$$n\left(t\right)=\frac{C_{b}C_{S_{0}}\sigma k_{a}}{C_{b}\sigma k_{a}+k_{d}}\left(1-\exp\left[-\left(C_{b}\sigma k_{a}+k_{d}\right)t\right]\right).$$

The concentration *C_b_*(*t*) obeys the diffusion equation with the initial condition *C_b_
*= *C*_0_ at *t* = 0:(3)$$\frac{\partial C_{b}}{\partial t}=D\frac{\partial^{2}C_{b}}{\partial x^{2}},$$
where *D* is the diffusivity of adsorbing particles, and coordinate *x* = 0 corresponds to the membrane surface. As follows from the model developed in [34], the solution to Equation (3) has the form
(4)$$C_{b}\left(x,t\right)=\int_{0}^{t}\varphi \left(\eta \right)\frac{x}{2\sqrt{\pi D{\left(t-\eta \right)}^{3}}}e^{-x^{2}/4D\left(t-\eta \right)}d\tau +\frac{2C_{0}}{\sqrt{\pi }}\int_{0}^{x/2\sqrt{Dt}}e^{-\xi^{2}}d\xi ,$$
where *φ* is the subsurface concentration of adsorbed substance, and *η* and *ξ* are variables of integration.

As follows from the insets in the graphs in Figure 6, the time dependences {[κ(*t* = 0) − κ(*t*)]/κ(*t* = 0)} × 100% are very well approximated by functions in the form of $Y_{0}\left(1-\exp\left[-t/\tau \right]\right).$ Obviously, at *t* → ∞, the dependence {[κ(*t* = 0) − κ(*t*)]/κ(*t* = 0)} × 100% reaches a stationary level, which corresponds to the adsorption–desorption equilibrium. Table 1 shows the values of pre-exponential factors *Y*_01_ (%) and relaxation times *τ*_1_ (min) for MB concentrations *C*_0_ = 0.015, 0.02, and 0.025 mg/mL for the solutions based on natural water, and Table 2 shows the same values *Y*_02_ and *τ*_2_ for the solutions based on DDW.

According to the data of Table 1 and Table 2, in the case of solutions based on natural water, the equilibrium (at *t* → ∞) amount of adsorbed MB (the quantity of MB removed from the solution), which is determined by the factor *A*, exceeds this value for the solutions based on DDW, that is, MB is most effectively removed from the solutions based on natural water. At the same time, the rate of MB decrease is determined by the value 1/*τ*_1_ or 1/*τ*_2_, i.e., this rate is higher for the solutions based on DDW. This was to be expected: when Nafion is soaked in the solution based on natural water, the polymer fibers are unwound, i.e., the surface area of the membrane, containing adsorption centers, increases substantially compared to the case of swelling in DDW (no unwinding effect). Within the framework of the model that the adsorption is controlled by diffusion (see Equations (3) and (4)), it can be argued that the diffusion coefficient *D* inside the layer of unwound polymer fibers is significantly less than the diffusion coefficient *D* in a free liquid. Unfortunately, we cannot perform a more detailed quantitative analysis of adsorption dynamics based only on the data in Table 1 and Table 2. Still, some numerical estimates can be made. Namely, for MB concentrations of 0.015, 0.02, and 0.025 mg/mL, the time ratios *τ*_1_/*τ*_2_ for solutions based on natural water and DDW are equal, respectively, to 2.67, 2.63, and 2.4, i.e., diffusion in a free liquid is accelerated by ~3 times compared with a layer containing unwound fibers.

Let us further consider at a qualitative level the desorption of water upon drying the membrane; the adsorption and desorption of water from the Nafion surface was considered in Refs. [36,37]. The adsorption of “MB + water” particles on the surface of Nafion can occur due to the fact that the surface of Nafion is negatively charged because of the dissociation of terminal sulfonic groups, which is accompanied by the transfer of protons into the bulk of liquid; see [1]. As shown in [38], the MB particle is polar in its ground state, while this particle in the excited state is nonpolar. At the same time, the water molecule is polar with the dipole moment *d* = 1.84 *D*, where *D* = 3.33 × 10^−30^ Q × m. Thus, it can be assumed that the “MB + water” particles form polar complexes with a dipole moment **d**′. The energy of the dipole interaction with the charged membrane can be presented as a scalar product—**d′E**, where **E** is the electric field strength vector near the Nafion surface.

Without accounting for the screening effects, the electric field strength of a uniformly charged surface is *E*_0_(*t*) = σ′(*t*)/(*εε*_0_), where σ′(*t*) is the surface charge density on the Nafion membrane, *ε* = 81 is the static permittivity of water, and *ε*_0_ = 0.9 × 10^−11^ F/m is the electrical constant; see [39]. Here, we assume that after removal of the Nafion plate from the solution, the polymer surface eventually becomes electrically neutral, i.e., the charge density σ′(*t*) → 0. Taking into account the effects of screening due to the presence of ions in liquid, the electric field strength has the form *E*(*x*,*t*) = *E*_0_(*t*)exp(−*x/R_D_*), i.e., this value decreases on the scale of the Debye screening radius *R_D_*; for a symmetrical monovalent electrolyte, we have for *R_D_:*(5)$$R_{D}=\sqrt{\frac{\varepsilon \varepsilon_{0}kT}{8\pi e^{2}n_{i0}}}{,}_{\;}$$
where *n_i_*_0_ is the equilibrium volume number density of ions, *x* is the coordinate perpendicular to the membrane surface (*x* = 0 corresponds to the position of the surface), *k* is the Boltzmann constant, and *e* is the elementary charge; see [40]. According to the results of measurements of the pH value close to the Nafion surface [41], this value at the distance *x* = 1–10 mm eventually reaches a stationary level: pH = 5.5. Apparently, it is precisely this pH value that corresponds to the equilibrium value *n_i_*_0_ in Equation (5). After finding *n_i_*_0_ and substituting it into Equation (5), we obtain that *R_D_
*≈ 100 nm. In addition, short-range dispersion forces act on the “MB + water” complexes from the polymer membrane. The potential of dispersion forces is −*A*/*x*^6^, where *A* is a dimensional constant, and the radius of dispersion forces is ~1 nm; see monograph [42]. Finally, the exchange repulsion potential *B/x*^12^ acts from the membrane surface, where *B* is another dimensional constant; see [42]. Thus, the “MB + water” complexes are located in the potential well
(6)$$W_{1}\left(x,t\right)=-A/x^{6}+B/x^{12}-C\left(t\right)\exp\left(-x/R_{D}\right),$$
where *C*(*t*) is a time-dependent dimensional parameter; the condition *C*(*t*) → 0 at *t* →∞ should be met. Thus, the last term in Equation (6) finally disappears, and the potential *W*_1_ acquires a well-known form, which is called the Lennard–Jones, L–J, or “6–12” potential
(7)$$W_{2}\left(x\right)=-A/x^{6}+B/x^{12}.$$

Consider the case *t* = 0 (the Nafion plate has just been removed from the liquid), i.e., we have, instead of formula (6), the following:(8)$$W_{1}\left(x,0\right)=-A/x^{6}+B/x^{12}-C\left(0\right)\exp\left(-x/R_{D}\right).$$

Let us find the positions *x*_0_′ and *x*_0_ of the minima of functions *W*_1_(*x*,0) and *W*_2_(*x*). In the latter case, after differentiating of Equation (7), we obtain
(9)$$x_{0}=\sqrt[{6}]{\frac{2B}{A}.}$$

As follows from the general principles of dispersion interactions (see [43]), the coordinate *x*_0_ (the minimum of the potential *W*_2_(*x*)) is related to the shortest distance between the centers of two interacting particles; these particles are assumed to be spherical, and these spheres are in contact with one another. In our case, these particles are the “MB + water” complex and the Nafion macromolecule, i.e., the coordinate *x*_0_ is about 1 nm. Let us now find the coordinate *x*_0_′, which corresponds to the minimum of *W*_1_(*x*,0). After differentiation in (8), we obtain the algebraic equation:(10)$$\frac{6A}{{\left(x_{0}^{\prime }\right)}^{7}}-\frac{12B}{{\left(x_{0}^{\prime }\right)}^{13}}+\frac{C\left(0\right)}{R_{D}}\exp\left(-\frac{x_{0}^{\prime }}{R_{D}}\right)=0.$$

This equation can be solved analytically by making a number of simplifications. First, let us assume that the electric field due to the charged surface should not lead to a significant change in *x*_0_′ as compared to *x*_0_. Thus, the condition *x*_0_′ << *R_D_* = 100 nm must be satisfied, that is, [*C*(0)/*R_D_*]exp(−*x*_0_′/*R_D_*) ≈ *C*(0)/*R_D_*. In addition, we assume that at *x* = *x*_0_′, the Coulomb interaction of a polar particle with a charged surface, which is described in (10) by the term *C*(0)/*R_D_*, exceeds essentially the dispersion interaction, which, as is known [42], is due to the fluctuations of dipole moments of particles. Therefore, we can put in (10) *A* = 0. After these simplifications, we arrive at
(11)$$x_{0}^{\prime }=\sqrt[{13}]{\frac{12BR_{D}}{C\left(0\right)}}.$$

Note that this solution is valid only providing that the surface is charged, i.e., the condition *C*(0) ≠ 0 should be met. Obviously, when the charge on the surface is neutralized, i.e., when *C*(0) = 0, the values *x*_0_ and *x*_0_′ must coincide. The curves *W*_2_(*x*) and *W*_1_(*x*,0) are qualitatively shown in Figure 10a,b, respectively. It is seen that the depth of the potential well decreases with time, cf., panels (a) and (b).

As the molar mass of MB is 319.85 g/mol, while the molar mass of water is 18 g/mol, we can assume that MB particles in the potential wells *W*_2_(*x*) and *W*_1_(*x*,0) are immobile, while water molecules should oscillate relative to the position of the MB particle; the area of these oscillations is marked with a double-arrowed horizontal straight line. The points of intersection of this line with the curves *W*_2_(*x*) and *W*_1_(*x*,0) set the height of the desorption thresholds Δ*W*_1_ and Δ*W*_2_ along the ordinate axis. It is seen that |Δ*W*_1_| > |Δ*W*_2_|, where |Δ*W*_1_| is related to panel (b) (*t* = 0, the membrane surface is charged), while |Δ*W*_2_|is related to panel (a) (the membrane surface is neutral). Following the general concepts of kinetic processes—see, for example, [43]—the desorption flux is proportional to the value exp(−Δ*W/kT*), i.e., at *t* = 0, the desorption of water is hindered. As |Δ*W*(*t*)| is a time-decreasing function, |Δ*W*(*t*)|→|Δ*W*_2_|, i.e., the water desorption flux must increase with time.

Thus, in the case of a flat membrane surface, the adsorption of water followed by desorption can apparently be described analytically, providing that the constants *A*, *B*, and *C*(*t*) are known to us. However, in the case of polymer fibers unwound into the bulk of liquid, the adsorption dynamics can no longer be described within the framework of such a simple model, as the electrically charged surface of the adsorbent is not flat, and the length of unwound (also charged) polymer fibers reaches 300 microns (see [19]), i.e., it substantially exceeds the value of *R_D_*. In this case, the problem of the distribution of the electric field strength *E* becomes very difficult, especially bearing in mind the nonstationary character of this problem. At present, we are developing an adequate theoretical model for describing adsorption and desorption with accounting for the unwinding of polymer fibers. 

However, we can assume that a decrease in the desorption threshold (decreasing of |Δ*W*_1_| up to the level of |Δ*W*_2_|) leads to the redshift in the line at wavelength λ = 646 nm to the line at wavelength λ = 666 nm (see the triplet in Figure 8a,b). Indeed, as the water molecule has a dipole moment, it is captured on a charged polymer surface due to Coulomb forces; in this case, the potential well *W*(*x*), inside which the water molecule performs a finite motion (oscillations), is deep enough; see Figure 10b. However, as the zero charge on the membrane is being restored, the Coulomb interaction between the water molecule and the Nafion surface disappears, and the potential well holding water molecules is only due to dispersion forces, which result from the fluctuations in the dipole moments of the interacting particles. Obviously, in this case, the depth of the potential well is much smaller than in the case of the Coulomb interaction, cf., Figure 10a,b. As the frequency of the oscillator is $\omega =\sqrt{\frac{k}{m}},$ where *m* is the effective mass and *k* is the elasticity constant, and the potential well for the oscillator (the potential energy for a quasi-elastic force) has the form *W*(*x*) = *kx*^2^/2, the value of *k* decreases upon reducing *W*(*x*); that is, when passing from the Coulomb to the dispersion interaction, the oscillator frequency *ω* should also decrease. Apparently, the triplet structure corresponds to the situation when the potential energies of the Coulomb and dispersion interactions are approximately the same.

In the process of drying of the membrane, which was soaked in the MB solution based on ordinary water, the transition from the three-component complex to the two-component complex is completed at *τ*_1_′ ≈ 12 min, and in the case of the MB solution based on DDW, this transition is completed at *τ*_2_′ ≈ 4 min. Obviously, when calculating the rates of water desorption, we can ignore the difference in molar masses for natural water and DDW, as the deuterium content in natural water is 157 ppm, while in DDW, the deuterium content is 1 ppm. The ratio *τ*_2_′/*τ*_1_′ = 3, i.e., there exists a very good correlation with the ratio of the times *τ*_2_/*τ*_1_ taken from Table 1 and Table 2; see above.

Concluding this section, we can claim that the unwinding of polymer fibers slows down the adsorption of MB on the Nafion surface, and also slows down the desorption of water molecules from this surface. As adsorption and desorption are controlled by diffusion, it can be taken as a zero approximation that the observed effects are caused by a decrease in the diffusion coefficient inside the layer of unwound polymer fibers. In our subsequent works, we will carry out numerical simulations of diffusion processes near a charged surface, taking into account the structure of the brush type of charged rods, directed normal to this surface.

## 5. Conclusions

The rate of MB adsorption on the Nafion surface depends on the isotopic composition of the MB aqueous solution, which is associated with the effect of unwinding polymer fibers from the membrane surface into the liquid bulk. This is due to the fact that adsorption is controlled by diffusion, and diffusion processes are slowed down inside the layer of unwound fibers, whereas the unwinding effect is controlled by the isotopic composition of liquid.Certain bands of the absorption spectrum of MB, adsorbed on the Nafion surface, are redshifted as compared to the same bands in the MB solution. The effect is associated with the formation of molecular complexes between MB, water, and Nafion. These complexes are formed due to short-range dispersion forces and long-range Coulomb forces; Coulomb forces are caused by the presence of negative charge on the membrane surface, resulting from the dissociation of terminal sulfonic groups and the transfer of protons into the liquid bulk. When the Nafion plate is removed from the liquid, the charge on the membrane surface is reduced to zero level, which leads to the disappearance of the Coulomb attraction and, accordingly, to a decrease in the desorption barrier for water molecules. The decrease in charge on the membrane surface is controlled by the diffusion kinetics. This is manifested in the behavior of the derivative *d*κ‘/*dt*. We can claim that in the case of polymer fibers, unwound into the bulk of liquid, the charge on the surface decreases slower than in the absence of the unwinding effect.We can claim that it is possible to control the dynamics of adsorption and desorption processes by infinitesimal changes in the deuterium content (from 3 to 157 ppm) in an aqueous solution, in which a polymer membrane swells.

## Figures and Tables

**Figure 1 polymers-14-04007-f001:**
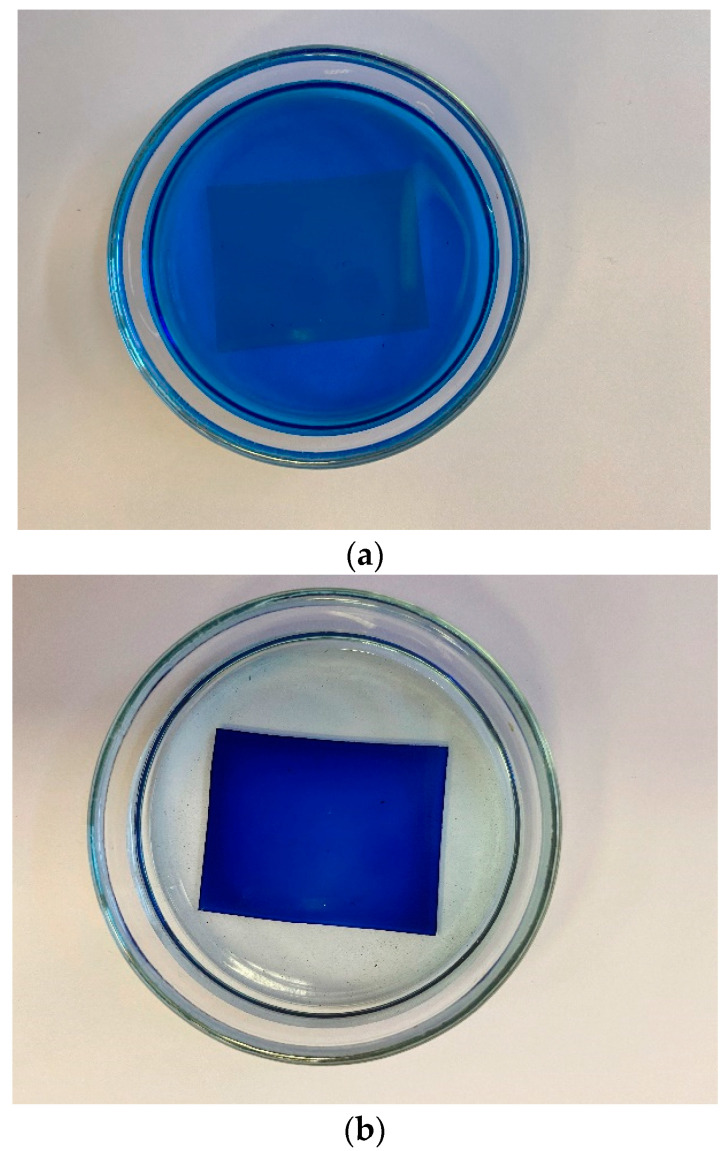
The process of soaking Nafion in the MB solution with concentration 0.015 mg/mL. Panel (**a**) corresponds to the start of soaking. Panel (**b**) corresponds to the end of soaking.

**Figure 2 polymers-14-04007-f002:**
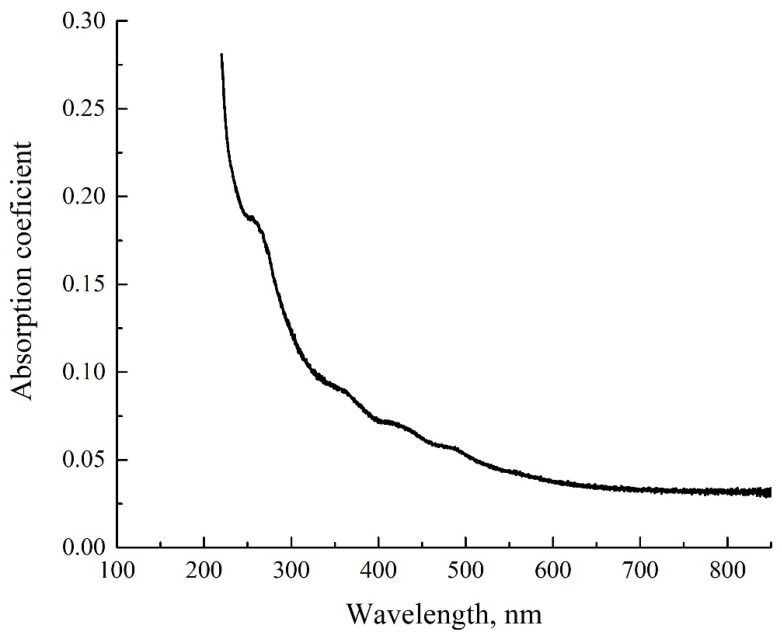
The absorptivity spectrum of dry Nafion.

**Figure 3 polymers-14-04007-f003:**
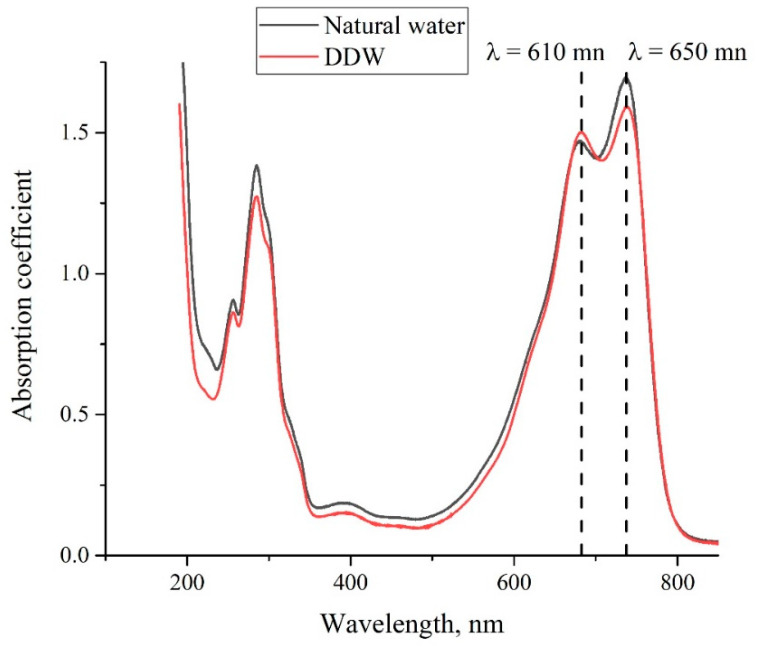
The absorptivity spectrum of the MB solution with concentration 0.015 mg/mL, based on natural water and DDW.

**Figure 4 polymers-14-04007-f004:**
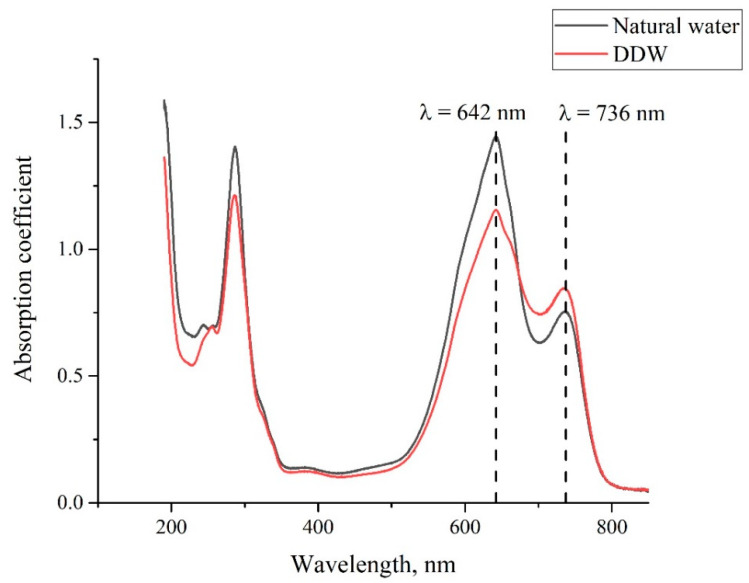
The absorptivity spectrum of Nafion membrane immediately after removing the plate out of the MB solution with weight concentration 0.015 mg/mL based on natural water, and DDW.

**Figure 5 polymers-14-04007-f005:**
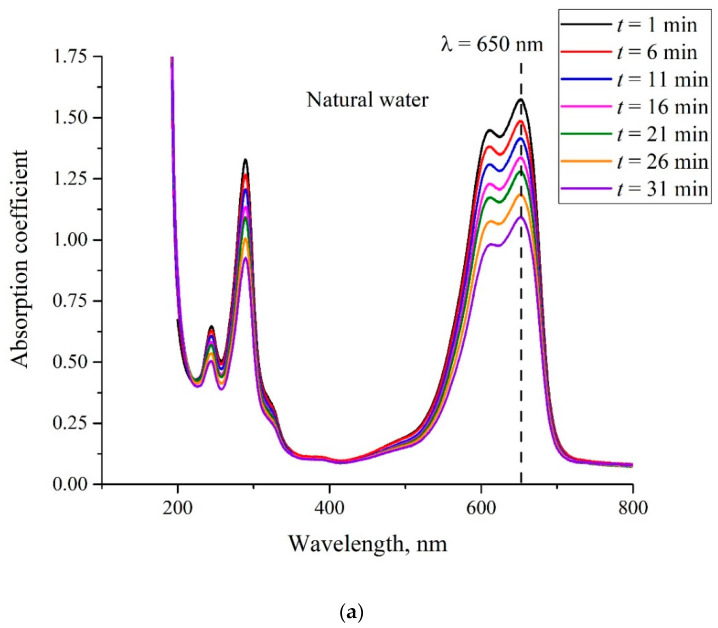
Absorption spectrum of the MB solution with concentration 0.015 mg/mL. The inset shows the time of soaking *t*. Panel (**a**)—the MB solution is based on natural water. Panel (**b**)—the MB solution is based on DDW.

**Figure 6 polymers-14-04007-f006:**
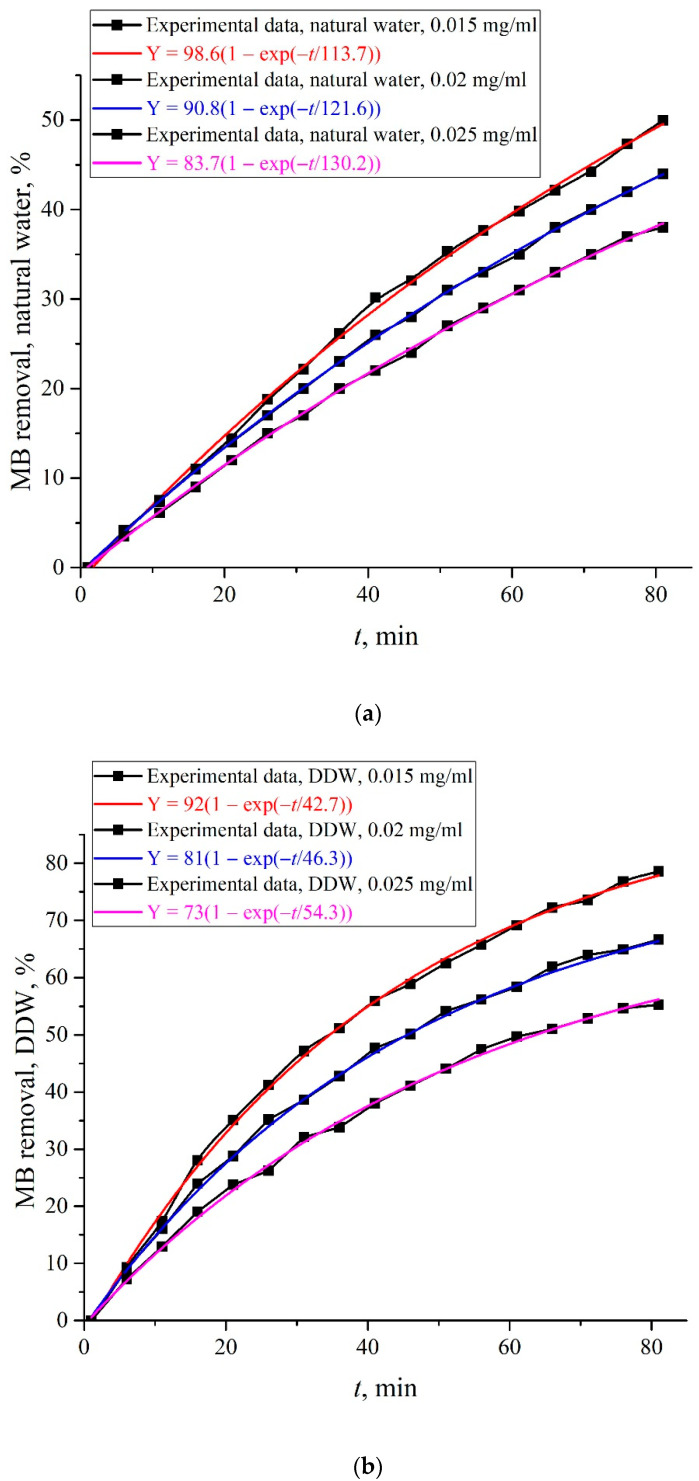
The dependences {[κ(*t* = 0) − κ(*t*)]/κ(*t* = 0)} × 100%. Panel (**a**)—the MB solution is based on natural water. Panel (**b**)—the MB solution is based on DDW.

**Figure 7 polymers-14-04007-f007:**
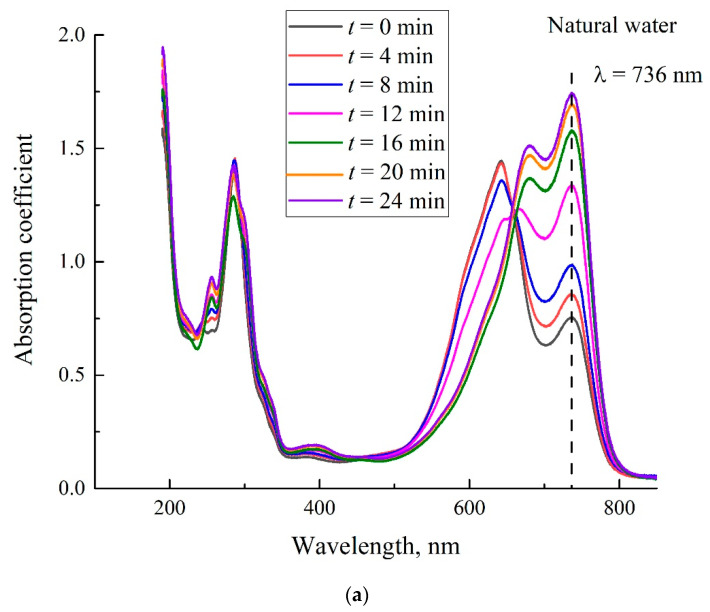
Absorptivity spectra of Nafion plate soaked in the MB solution with concentration 0.015 mg/mL. The inset shows the soaking time *t*. Panel (**a**)—the MB solution is based on natural water. Panel (**b**)—the MB solution is based on DDW.

**Figure 8 polymers-14-04007-f008:**
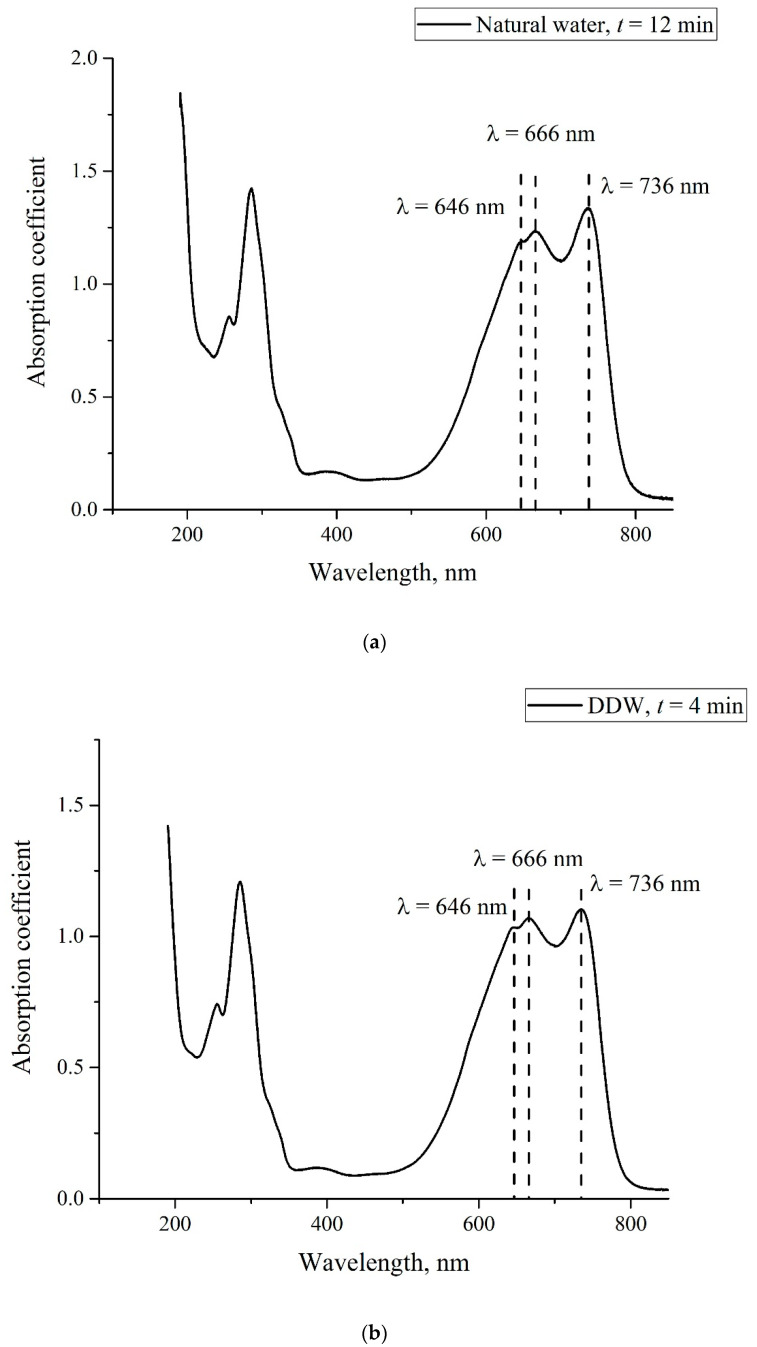
The triplet structures of the absorptivity spectra. Panel (**a**)—the MB solution with concentration 0.015 mg/mL is based on natural water, *t* = 12 min. Panel (**b**)—the MB solution with concentration 0.015 mg/mL is based on DDW, *t* = 4 min.

**Figure 9 polymers-14-04007-f009:**
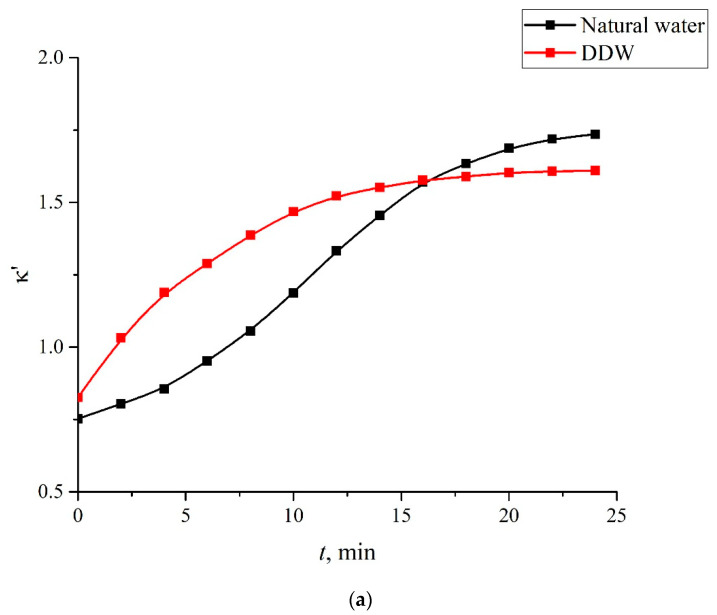
Dynamics of drying the Nafion plate after soaking in the MB solution based on natural water (black curves) and DDW (red curves). Panel (**a**)—dependences κ‘(*t*). Panel (**b**)—dependences *d*κ‘/*dt*.

**Figure 10 polymers-14-04007-f010:**
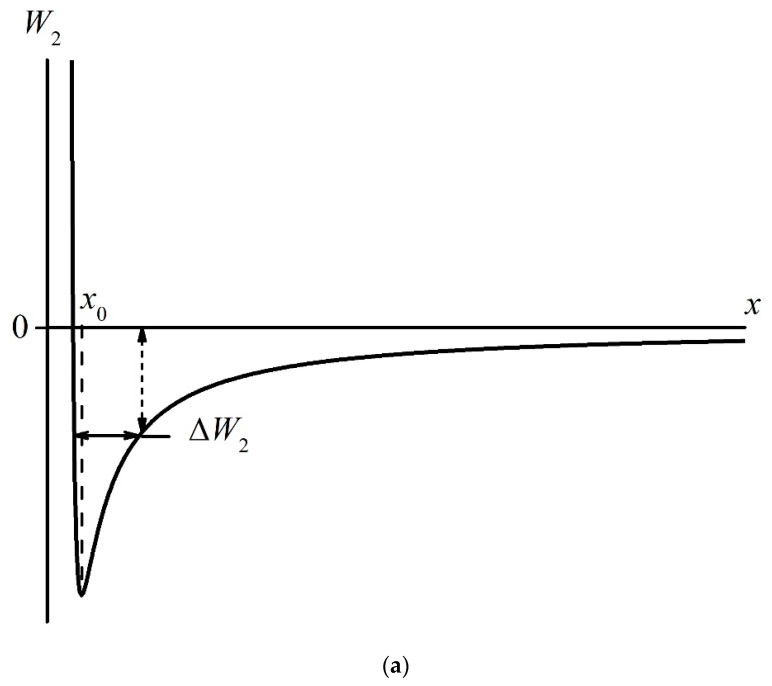
Potential energy of “MB + water + Nafion” complexes near the surface of Nafion. Panel (**a**) is related to the case *C*(*t*) = 0 (Equation (7), *W*_2_(*x*)). Panel (**b**) is related to the case *t* = 0 (Equation (8), *W*_1_(*x*,0)).

**Table 1 polymers-14-04007-t001:** The values of pre-exponential factors *Y*_01_ and relaxation times τ_1_ for natural water.

*C*_0_, mg/mL	*Y*_01_*,* %	*τ*_1_, min
0.015	98.6	113.7
0.02	90.8	121.6
0.025	83.7	130.2

**Table 2 polymers-14-04007-t002:** The values of pre-exponential factors *Y*_02_ and relaxation times *τ*_2_ for DDW.

*C*, mg/mL	*Y*_02_*,* %	*τ*_2_, min
0.015	92	42.7
0.02	81	46.3
0.025	73	54.3

## Data Availability

The data presented in this study are available on request from the corresponding author.

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
