# Peer review of "Adsorption of Methylene Blue on the Surface of Polymer Membrane; Dependence on the Isotopic Composition of Liquid Matrix"

_polymers, 2022, doi:10.3390/polym14194007_

Round 1

Reviewer 1 Report

This study is interesting. However, the following comments should be considered before this can be published.

1. The Introduction section seems to be too long. Can the authors make it concise and more straightforward highlighting the novelty of the current study? 

2. In the Introduction section the chemical structure of Nafion can be shown where the description is given. 

3. "In this case, a decolorization of the MB solution was observed. " (Line 181) It can be seen if the authors include two images i.e., initial and final images from the soaking stage.

4. Figures 3 (a) and (b) and Figure 4(a) and (b) should be merged into figures 3 (a) and (b) for better visualization and comparison.

5. Can the authors provide some detailed explanation about the triplet band after the 12th and 4th minutes for water and DDW solvents, respectively?

6. This study includes a lot of equations with various parameters. A nomenclature section can be added. 

7. The data related to the other concentrations of methylene blue solution can be provided in the supplementary section.

8. The conclusion section should be modified with the present study's findings. 

Author Response

The Reviewer is thanked for the constructive feedback. We have improved the text in accordance with the Reviewer’ remarks. Our responds to the Reviewer’ criticism are highlighted by italic font.

  1. The Introduction section seems to be too long. Can the authors make it concise and more straightforward highlighting the novelty of the current study? 

The Introduction Section has been significantly shortened, and the general statement of the problem has been substantiated in more detail.

  1. In the Introduction section the chemical structure of Nafion can be shown where the description is given. 

Thank you for this remark. The chemical structure of Nafion is shown in the Introduction Section.

  1. "In this case, a decolorization of the MB solution was observed. " (Line 181) It can be seen if the authors include two images i.e., initial and final images from the soaking stage.

Photographs showing the start and end of adsorption are shown in Fig. 1 (a) and (b).

  1. Figures 3 (a) and (b) and Figure 4(a) and (b) should be merged into figures 3 (a) and (b) for better visualization and comparison.

It was done in the new version of manuscript.

  1. Can the authors provide some detailed explanation about the triplet band after the 12th and 4th minutes for water and DDW solvents, respectively?

In the new version, we have proposed the following qualitative model. The surface of Nafion in water is negatively charged due to the dissociation of terminal sulfo groups and the release of free protons into the bulk of the liquid. When Nafion is removed from the water, the membrane surface becomes neutral again. As the membrane dries, water molecules are desorbed from the surface. Since the water molecule has a dipole moment, it is held on a charged surface due to Coulomb forces; in this case, the potential well, inside which the water molecule performs a finite motion (oscillations), is deep enough. However, as the zero charge on the membrane is restored, the Coulomb interaction between the water molecule and the Nafion surface disappears, and the potential well holding water molecules is due to dispersion forces that arise due to fluctuations in the dipole moments of the interacting particles. Obviously, in this case the depth of the potential well is much smaller than in the case of the Coulomb interaction. Since the frequency of the oscillator is proportional to the square root of the spring constant, and the potential well for the oscillator has the form W(x) = kx^2/2, then as W(x) decreases, the value of k also decreases, that is, when passing from the Coulomb to the dispersion interaction, the frequency of the oscillator should also decrease. Apparently, the triplet structure corresponds to the situation when the potential energies of the Coulomb and dispersion interactions are approximately the same.

  1. This study includes a lot of equations with various parameters. A nomenclature section can be added. 

This Section was added to the final version.

  1. The data related to the other concentrations of methylene blue solution can be provided in the supplementary section.

We have added concentration data for 0.015, 0.02 and 0.025 mg/mL to the new version. The results for all studied concentrations are qualitatively very similar, which is specially emphasized in the new version. The new version also provides a qualitative interpretation of the results obtained for different concentrations.

  1. The conclusion section should be modified with the present study's findings. 

This section has been completely rewritten. Of course, the main result is that it appears possible to control the dynamics of adsorption and desorption processes by infinitesimal changes in the deuterium content (from 3 to 157 ppm) in an aqueous solution, in which a polymer membrane swells.

Reviewer 2 Report

The manuscript entitled "Adsorption of Methylene Blue on The Surface of Polymer Membrane; Dependence on The Isotopic Composition of Liquid Matrix" has been investigated in details. The topic addressed in the manuscript is potentially interesting and the manuscript contains some practical meanings, however, there are some concerns which should be addressed by the authors for further proceeding:

1.       In the first place, I would encourage the authors to revise the abstract by reporting the key results. The "Abstract" section can be made much more impressive by highlighting your contributions. The contribution of the study should be explained simply and clearly.

2.       The readability and presentation of the study should be further improved.

3.       The Introduction section needs a major revision in terms of providing more accurate and informative literature review and the pros and cons of the available approaches and how the proposed method is different comparatively. Also, the motivation and contribution should be stated more clearly.

4.       The importance of the design carried out in this manuscript can be explained better than other important studies published in this field. I recommend the authors to review other recently developed works.

5.       More details and discussion regarding the different applications of polymeric materials should be reported in the draft.

6.       The performance of the proposed method should be better analyzed, commented and visualized in the experimental and analytical approaches.

7.       What makes the proposed method suitable for this unique task? What new development to the proposed method have the authors added (compared to the existing experimental and theoretical approaches)? These points should be clarified.

8.       All variables should clearly to be specified in the draft accompanied with their units.

9.       "Discussion" section should be edited in a more highlighting, argumentative way. The author should analysis the reason why the tested results is achieved.

10.   A comparison with some other available approaches in the literature should be added to the study.

11.   Figures should be redesigned and presented in the high quality format.

12.   The authors should clearly emphasize the contribution of the study. Please note that the up-to-date of references will contribute to the up-to-date of your manuscript. There are several out of date and or irrelevant cited work. Accordingly, literature review shall be completely shorten and revised. The studies named- Samsudin et al., Polymers 2022, 14(17), 3565; Tsioptsias et al., Polymers 2022, 14(16), 3434.

Author Response

The Reviewer is thanked for the constructive feedback. We have improved the text in accordance with the Reviewer’ remarks. Our responds to the Reviewer’ criticism are highlighted by italic font.

  1. In the first place, I would encourage the authors to revise the abstract by reporting the key results. The "Abstract" section can be made much more impressive by highlighting your contributions. The contribution of the study should be explained simply and clearly.

In accordance with the advice of the reviewer, we have completely rewritten the abstract. We agree with the reviewer that there were no specific statements in the first version of the abstract.

  1. The readability and presentation of the study should be further improved.

In our opinion, in the new version we have improved the readability and quality of the presentation. On the other hand, we only had 10 days to improve the text, and besides, there are no native speakers in our team, so it is a big problem for us to improve the quality of the text in accordance with the comments of English-speaking readers.

  1. The Introduction section needs a major revision in terms of providing more accurate and informative literature review and the pros and cons of the available approaches and how the proposed method is different comparatively. Also, the motivation and contribution should be stated more clearly.

We have significantly shortened the Introduction section and also reduced the total number of citations. In addition, in the new version, we refer to works on measuring the dynamics of MB adsorption using the spectrophotometric technique (references [57 - 62] in the new version); these works were published in 2020 - 2022. Thus, the method proposed by us significantly develops the already existing techniques, taking into account the fact that we study the adsorption of MB considering the isotopic composition of the aqueous solution.

  1. The importance of the design carried out in this manuscript can be explained better than other important studies published in this field. I recommend the authors to review other recently developed works.

       As we noted in the previous paragraph, we completely rewrote the Introduction section, shortening this section and significantly reducing the number of cited literature. In addition, we have removed the references to works that have lost their relevance, leaving only references to works that have been released most recently. Finally, we have added references to the most recent works on the study of the dynamics of MB adsorption on various adsorbents using spectrophotometry.

  1. More details and discussion regarding the different applications of polymeric materials should be reported in the draft.

      This remark is not very clear to us. It was possible to write an overview of various applications of polymeric materials in the Introduction, but this would lead to a significant increase in the volume of the manuscript, especially the Introduction section, which the reviewer recommends shortening. Of course, in the new version, we consider various applications of polymeric materials, taking into account the adsorption of various external particles on the polymer surface. Our manuscript is devoted to the influence of the isotopic composition on the dynamics of adsorption on the surface of Nafion. Since these works were started in our group, we could refer to our own articles, but we did not do this for ethical reasons.

  1. The performance of the proposed method should be better analyzed, commented and visualized in the experimental and analytical approaches.

       In the new version, we have presented data for other MB concentrations. Based on the theoretical approximation of the experimental dependences, numerical estimates were made and a conclusion was made about the effectiveness of the proposed technique for studying the dynamics of adsorption with allowance for the isotopic composition of the liquid.

  1. What makes the proposed method suitable for this unique task? What new development to the proposed method have the authors added (compared to the existing experimental and theoretical approaches)? These points should be clarified.

As already noted in the previous paragraphs, we do not claim the uniqueness of our methodology. Indeed, the technique we use is widely applied, and we refer to the relevant works published most recently. However, in our opinion, our technique is effective precisely for studying the effect of the isotope composition on the adsorption dynamics. Using this technique allowed us to conclude that it is possible to control the dynamics of adsorption and desorption processes by infinitesimal changes in the deuterium content (from 3 to 157 ppm) in an aqueous solution, in which a polymer membrane swells

  1. All variables should clearly to be specified in the draft accompanied with their units.

      We agree with this remark. Therefore, we have added a nomenclature section. At the same time, we do not indicate the units of measurement of the variables used, since we do not make numerical estimates. Unfortunately, the values of the main parameters that need to be known for numerical calculations are not known to us, and we specially note this in the new version.

  1. "Discussion" section should be edited in a more highlighting, argumentative way. The author should analysis the reason why the tested results is achieved.

      The Discussion section has been completely rewritten. In particular, a semi-quantitative analysis of the appearance of a triplet structure, taking into account the effect of unwinding of polymer fibers, was added to this section.

  1. A comparison with some other available approaches in the literature should be added to the study.

      We would like to emphasize once again that our work is devoted to the effect of the isotopic composition on the dynamics of adsorption. We are not aware of any work where this problem has been studied, so we simply cannot provide any comparative analysis. In our work, we used a technique based on spectrophotometry; this technique is widely used, and we refer to the most recent works carried out with using this technique.

  1. Figures should be redesigned and presented in the high quality format.

       We agree with this remark. The quality of the drawings has been significantly improved. In addition, Fig. 1 (b) was added, Figs. 3 (a) and (b), as well as Figs. 4(a) and (b), were merged. In addition, in Fig. 6(a) and (b) the plots for other MB concentrations were added.

  1. The authors should clearly emphasize the contribution of the study. Please note that the up-to-date of references will contribute to the up-to-date of your manuscript. There are several out of date and or irrelevant cited work. Accordingly, literature review shall be completely shorten and revised. The studies named- Samsudin et al., Polymers 2022, 14(17), 3565; Tsioptsias et al., Polymers 2022, 14(16), 3434.

We agree with this remark. Only the most relevant references were left in the literature review, and now the number of cited literature is 75 references. We are grateful to the referee for the works mentioned here. These works are very interesting, and we refer to them (references [16] and [17]).

Round 2

Reviewer 2 Report

Unfortunately I should declare, Author revised the draft in a disappointing way!!! They almost ignored all concerns and comments, also they added more than 30 new references, mostly irrelevant and self-citation. Accordingly, I recommend once again they revise the draft in a major-scientific way, otherwise the paper will not-recommend for possible publication. 

Author Response

We are grateful to the referee for carefully reading the new version of the manuscript. We read with some surprise that we have added over 30 new citations. In fact, the original version had 100 citations, and the next version had 75 citations. We added 5 new citations to the most recent (released in 2020-2022) experimental works on the study of MB adsorption on various adsorbents. On the whole, the results obtained by us do not contradict the data presented in these works, that is, citing these works seems quite reasonable.

In the most recent version of the manuscript, we again reduced the number of citations; this number is now 43. We have also reduced the number of self-citations, leaving only our articles published in 2022. The number of self-citations in the most recent version is 6, and the total self-citation rate is 13.9%. In addition, we have shortened the Introduction section again. Let me remind you that this section provides a justification for conducting experiments on the topic we have chosen, taking into account the results we obtained and the results obtained by other authors. In the Introduction, we wrote why we used MB solutions based on natural water and deuterium-depleted water. In addition, in the Introduction, we substantiated why, among other antioxidants, we studied specifically the adsorption of MB on the surface of Nafion. This is due to the widespread use of MB in the treatment of COVID. Since our special issue is devoted, among other things, to the applications of polymeric membranes in biology, citing these works seems quite appropriate.

We would like to specially note that no one before us has studied adsorption from aqueous solutions with different deuterium contents, since, it would seem, the presence / absence of deuterium should not influence on adsorption processes. It turned out however that the isotope effects manifest themselves at very small changes in the deuterium content in water - from 3 to 157 ppm, that is, it becomes possible to control adsorption processes by changing the isotopic composition. In addition, the dynamics of water desorption from the surface of Nafion, which swelled in aqueous solutions of MB with different deuterium contents, was studied. It turned out that isotope effects also manifest themselves in desorption processes. To the best of our knowledge, these results are new and have not been published anywhere before.

We hope that the most recent version of the manuscript contains the optimal number of relevant citations.